# A Survey on Future Frame Synthesis: Bridging Deterministic and Generative Approaches

## Abstract

Future Frame Synthesis (FFS) aims to enable models to generate sequences of future frames based on existing content. This survey comprehensively reviews historical and contemporary works in FFS, including widely used datasets and algorithms. It scrutinizes the challenges and the evolving landscape of FFS within computer vision, with a focus on the transition from deterministic to generative synthesis methodologies. Our taxonomy highlights the significant advancements and shifts in approach, underscoring the growing importance of generative models in achieving realistic and diverse future frame predictions.

## 1 Introduction

The goal of the future frame synthesis (FFS) task is to generate future frames based on a sequence of historical frames (Srivastava et al., 2015) or just a single context frame (Xue et al., 2016), with or without additional control signals. The learning objective of this FFS is also considered to be key to building a world model (Ha & Schmidhuber, 2018; Hafner et al., 2023b). FFS is closely related to low-level computer vision processing techniques, particularly when synthesizing near frames (Liu et al., 2017; Wu et al., 2022b; Hu et al., 2023b). However, FFS diverges from other low-level tasks by implicitly demanding a more complex understanding of scene dynamics and temporal coherence, which is often characteristic of high-level vision tasks. The challenge lies in designing models that can achieve this balance efficiently, using a moderate number of parameters to minimize inference latency and resource consumption, thereby making FFS suitable for real-world applications. This unique position of FFS demonstrates its integral role in bridging the gap between low-level perception & prediction and high-level understanding & generation within computer vision.

Current FFS algorithms can generally be divided into two categories. One category entails **referencing** pixels from the current frames (typically the last observed frame) to construct future frames. However, this group of methods inherently confronts difficulties in modeling objects' appearance and disappearance (birth and death) within the scene. A shared feature of these methods is that they can make highly accurate predictions in the short term, but their accuracy diminishes over longer periods. This kind of research is also commonly referred to as video prediction (Oprea et al., 2020). The other category encompasses methods that involve **generating** new frames from scratch. Although these approaches offer the promise of capturing the birth-and-death phenomena in object dynamics, they predominantly focus on modeling pixel-level distributions. As a result, they often lack an integrated understanding of the underlying real-world context, which is crucial for creative synthesis capabilities.

Before our work, two reviews in 2020 on video prediction (Oprea et al., 2020; Rasouli, 2020a) cover a lot of early technical content. Recently, there have also been literature reviews on "text-to-video generative model" (Liu et al., 2024) and "long video generation" (Li et al., 2024; Sun et al., 2024b). Our survey mainly highlights the latest advancements and the connections between predictive and generative methodologies. We believe that the future of long-term FFS hinges on a synergistic integration of prediction and generation techniques. This unified approach would combine contextual constraints with enhanced comprehension, addressing challenges within a cohesive framework.

As background, we will introduce the problem, the dataset, and the fundamental challenges faced in Section 2. Our taxonomy focuses on the stochasticity of methods. In Section 3, we introduce deterministic algorithms

that aim to perform pixel-level fitting based on deterministic target frames. However, due to pixel-level metrics encouraging models to average over multiple equally probable outcomes, they often produce blurred outputs. In Section 4, we discuss algorithms that attempt to imbue models with the ability to make stochastic predictions in motion. This includes methods that introduce stochastic variables or distributions into deterministic models and approaches that directly utilize probabilistic models. Such algorithms allow models to sample from motion distributions, encouraging the generation of predictions that deviate significantly from the target frame but remain reasonable. Given the limited creativity of existing FFS algorithms for high-resolution natural video data, which struggle with creating future frame sequences containing many birth-and-death phenomena, we introduce the generative FFS task in Section 5, which prioritizes generating reasonable video frame sequences in extended temporal durations over pixel-level accuracy. In Section 6, we explore the diverse applications of FFS across various domains, highlighting its significance in autonomous driving, robot navigation, the cinema industry, meteorology, and anomaly detection. These applications underscore the importance and potential of FFS as a tool for understanding and interacting with the world around us. In Section 7, we provide an overview of previous surveys on video prediction and video diffusion models. We also clarify the focus of our survey, which comprehensively reviews historical and contemporary works in FFS, focusing on the transition from deterministic to generative synthesis methodologies. It emphasizes the growing importance of generative models in achieving realistic and diverse future frame predictions, highlighting significant advancements and shifts in approach.

## 2 Future frame synthesis

### 2.1 Problem Definition

The nature of FFS task involves forecasting future frames or sequences in a video based on the analysis of past frames. The primary goal is to develop models that can accurately anticipate the visual content and probable motion of subsequent frames in a video sequence. This can be formulated as a conditional generative modeling problem, where given a sequence of observed frames $X_{t_1:t_2}$, the goal is to synthesize the future frames $Y_{t_2+1:t_3}$.

$$Y_{t_2+1:t_3} = X_{t_1:t_2} \cdot P(Y_{t_2+1:t_3}|X_{t_1:t_2}). \tag{1}$$

In Eq. (1), $t_1$ represents the initial time step, $t_2$ is the final time step for observed frames, and $t_3$ denotes the last time step for synthesizing future frames. The challenge lies in learning a mapping function that captures the complex spatiotemporal dependencies within the video sequence.

Many FFS algorithms also make use of additional information, which can be auxiliary data $A_{t_1:t_2}$ from videos, such as depth maps, landmarks, bounding boxes, and segmentation maps, to help the model understand the video. It can also include human input control signals $C_{t_2+1:t_3}$, such as text instructions and strokes, with the goal of enabling the model to generate future video frame sequences based on specific future trajectories. Based on this consideration, we can extend the formula of the FFS task to a more comprehensive version as Eq. (2):

$$Y_{t_2+1:t_3} = X_{t_1:t_2} \cdot P(Y_{t_2+1:t_3}|X_{t_1:t_2}, A_{t_1:t_2}, C_{t_2+1:t_3}). \tag{2}$$

### 2.2 Datasets

The advancement in video synthesis models is greatly dependent on the diversity, quality, and characteristics of training datasets. A general observation is the varying suitability of datasets based on their dimensionality and size, where lower-dimensional datasets, typically with smaller data sizes, may suffer from limited generalizability. In contrast, higher-dimensional datasets provide a broader range of data, contributing to stronger generalization capabilities in the models. We offer an overview of the most widely used datasets in video synthesis in Table 1, highlighting their amounts of data and additional supervisory modalities, thereby providing a comprehensive picture of the current dataset landscape in this field. In cases where specific

| Dataset | Category | # Videos | # Clip Frames | Resolution | Extra Annotations |
|---|---|---|---|---|---|
| KTH Action (Schuldt et al., 2004) | Human | 2, 391 | 95* | $160 \times 120$ | Class |
| Caltech Pedestrian (Dollar et al., 2011) | Human | 137 | 1, 824 | $640 \times 480$ | Bounding Box |
| HMDB51 (Kuehne et al., 2011) | Human | 6, 766 | 93* | $414 \times 404$† | Class |
| UCF101 (Soomro et al., 2012) | Human | 13, 320 | 187* | $320 \times 240$ | Class |
| J-HMDB (Jhuang et al., 2013) | Human | 928 | 34* | $320 \times 240$ | OF, Ins, HJ, Class |
| KITTI (Geiger et al., 2013) | Traffic | 151 | 323* | $1242 \times 375$ | OF, BBox, Sem, Ins, Depth |
| Penn Action (Zhang et al., 2013) | Human | 2, 326 | 70* | $480 \times 270$† | Human Joint, Class |
| SJTU 4K (Song et al., 2013) | General | 15 | 300 | $3840 \times 2160$ | - |
| Sports-1M (Karpathy et al., 2014) | Human | 1, 133, 158 | variable | variable | Class |
| Moving MNIST (Srivastava et al., 2015) | Simulation | 10, 000 | 20 | $64 \times 64$ | - |
| Cityscapes (Cordts et al., 2016) | Traffic | 46 | 869* | $2048 \times 1024$ | Semantic, Instance, Depth |
| YouTube-8M (Abu-El-Haija et al., 2016) | General | 8, 200, 000 | variable | variable | Class |
| Robotic Pushing (Finn et al., 2016) | Robot | 59, 000 | 25* | $640 \times 512$ | Class |
| DAVIS17 (Pont-Tuset et al., 2017) | General | 150 | 73* | $3840 \times 2026$† | Semantic |
| Something-Something (Goyal et al., 2017) | Object | 220, 847 | 45 | $427 \times 240$† | Text |
| ShapeStacks (Groth et al., 2018) | Simulation | 36, 000 | 16 | $224 \times 224$ | Semantic |
| SM-MNIST (Denton & Fergus, 2018) | Simulation | customize | customize | $64 \times 64$ | - |
| D$^2$-City (Che et al., 2019) | Traffic | 11, 211 | 750* | 1080p / 720p | BBox |
| Kinetics-700 (Carreira et al., 2019) | Human | 650, 000 | 250* | variable | Class |
| RoboNet (Dasari et al., 2019) | Robot | 161, 000 | 93* | $64 \times 48$ | - |
| Vimeo-90K (Xue et al., 2019) | General | 91, 701 | 7 | $448 \times 256$ | - |
| BDD100K (Yu et al., 2020) | Traffic | 100, 000 | 1175* | $1280 \times 720$ | BBox, Semantic, Depth |
| nuScenes (Caesar et al., 2020) | Traffic | 1, 000 | 40* | $1600 \times 900$ | BBox, Semantic |
| WebVid (Bain et al., 2021) | General | 10, 732, 607 | 449* | $596 \times 336$ | Text |
| X4K1000FPS (Sim et al., 2021) | General | 4, 408 | 65* | $4096 \times 2160$ | - |
| SportsSlomo (Chen & Jiang, 2024) | Human | 130, 000 | 7 | $1280 \times 720$ | - |
| InternVideo2 (Wang et al., 2024b) | General | 2, 000, 000 | variable | variable | Text, Action |
| OpenDV-YouTube (Yang et al., 2024) | Traffic | 2, 139 | 100,000+* | variable | Text |

* denotes the mean value. † denotes the median value.

Table 1: Summary of the most used video prediction datasets, including the total number of videos, frame count for each video clip, image resolution, and additional annotations, etc. (**OF**: Optical Flow, **BBox**: Bounding Box, **Sem**: Semantic, **Ins**: Instance, **HJ**: Human Joints)

dataset details are not reported either in the original paper or the project page, we calculate the mean or median statistics to maintain consistency in our analysis.

**Challenges.** 1. Unify the organization of image and video data. A substantial amount of computer vision work is conducted on the image modality. Therefore, image datasets may undergo more meticulous curation and possess more annotated data. The most commonly used billions-level image datasets include YFCC100M (Thomee et al., 2016), WIT400M (Radford et al., 2021) and LAION400M (Schuhmann et al., 2021). Given the vast amount of data involved, researchers may need to effectively leverage knowledge from foundational image models. When incorporating video data into training, we may need to filter out low-quality segments and select an appropriate sampling frame rate.

2. Determine the proportion of data from different domains. Computer graphics composite data, 2D anime data, real videos, and videos with special effects can have vastly different appearances. Moreover, we may not be able to unify data from different sources into a specific resolution because videos may have different aspect ratios, and details that only make sense at high resolutions (such as subtitles, and textures). Many frame synthesis methods are sensitive to resolution because of the correlation between resolution and object motion intensity (Sim et al., 2021; Hu et al., 2023b; Yoon et al., 2024).

## 2.3 Overall Challenges

There exist longstanding challenges in FFS field, including the need for algorithms that balance low-level pixel processing with high-level scene dynamics understanding, the inadequacy of perceptual and stochastic evaluation metrics, the difficulty in achieving long-term synthesis, and limited high-resolution dataset quality for stochastic motion and birth-and-death phenomena. This section provides an overview of challenges.

**Evaluation metrics.** The low-level metrics such as Peak Signal-to-Noise Ratio (PSNR) and Structural Similarity Index (SSIM), can only assess the pixel-wise accuracy of predicted pixels. To chase these metrics, researchers typically train models utilizing either the $l_1$ or $l_2$ loss function of pixel space. The models are prone to average over multiple plausible outcomes, often resulting in blurry predictions, known as the perception-distortion tradeoff (Blau & Michaeli, 2018). They tend to favor blurry predictions that nearly accommodate the ground truth over sharper and more plausible but imperfect generations that do not match the ground truth. Many researchers are increasingly looking towards alternatives, such as perceptual metrics (DeePSiM (Dosovitskiy & Brox, 2016), LPIPS (Zhang et al., 2018)), and stochastic metrics (IS (Salimans et al., 2016), FID (Heusel et al., 2017)). These metrics may better align with human perception. But even classifiers trained on human perception annotations have a relatively low level of agreement with humans in judging image quality (Kumar et al., 2022).

Visual researchers typically evaluate models based on the visual quality of the generated results. However, for many specific applications, we are not sure that visual quality is crucial. For instance, Dreamer-V3 (Hafner et al., 2023a) and VPT (Baker et al., 2022) have successfully built visual models on low-resolution frame sequences. Moreover, most research work on visual representations is built on relatively small image resolutions (Radford et al., 2021; He et al., 2022). We are concerned that an excessive pursuit of visual quality may lead us to favor models that focus on low-level features. In addition to matching human subjective perception, we also need to design metrics that evaluate a model's ability to capture scene dynamics and temporal variations.

Even with better evaluation metrics, optimizing for them is a challenge. During model training, researchers often use Imagenet classifiers as feature extractors (Johnson et al., 2016; Kumar et al., 2022) in comparing the generated results with the ground truth to optimize for both low-level and high-level features. There are also some GAN-based loss functions that can improve the image quality of the generated results (Huang et al., 2017; Zhang et al., 2020).

**Long-term synthesis.** While short-term video prediction has seen significant advancements, accurately synthesizing events over extended time horizons remains challenging because of long-term dependencies and complex interactions between objects in dynamic scenes. Iteratively using a short-term video prediction model will quickly lead to degraded results (Wu et al., 2022b; Hu et al., 2023b). Due to the inadequacy of model scale in forming a comprehensive understanding of the real world, most existing video synthesis models primarily model changes in pixel distribution. When faced with natural videos spanning extended periods, these models struggle to accurately predict object movements while maintaining visual quality. A promising approach for enhancing long-term synthesis capabilities is the incorporation of high-level structure information (Villegas et al., 2017). Understanding and utilizing higher-order information can enable the model to retain more key details, thereby achieving internal consistency within the video over longer time scales.

**Generalization.** The interplay between data volume and the complexity of models jointly determines the upper bounds of an algorithm's performance. Despite the vastness of video data available on the internet, the scarcity of high-quality video datasets suitable for video synthesis remains a limiting factor, and existing datasets also pose various challenges, such as simplistic data distribution, low resolution, and small motion scales. These issues make it difficult for video synthesis models to handle high-resolution content and large motion scales, thus restricting their practical utility to diverse and unseen scenarios. Achieving high-resolution video synthesis is a complex task demanding significant computational resources (Blattmann et al., 2023b). The intricacies involved make real-time applications challenging due to the substantial computational burden.

## 3 Deterministic synthesis

### 3.1 Raw Pixel Space

In short-term FFS, methods in raw pixel space have achieved good results. We hope to review existing methods and discuss the challenges faced.

### 3.1.1 Recurrent networks.

The exploration of recurrent neural networks in video synthesis is pioneered by PredNet (Lotter et al., 2016), which draws inspiration from predictive coding in neuroscience and employs a recurrent convolutional network for the effective processing of video features. Building on this, PredRNN (Wang et al., 2017) introduces significant enhancements by modifying Long Short-term Memory (LSTM) with a dual memory structure, aiming for enhanced spatiotemporal modeling. Despite its advancements, it encounters challenges with gradient vanishing in video synthesis tasks. Addressing such limitations, ConvLSTM (Shi et al., 2015) emerges as a pivotal model, ingeniously integrating LSTM with Convolutional Neural Network (CNN) to proficiently capture motion and spatiotemporal dynamics, a development that has significantly influenced subsequent video synthesis models. Further advancing the field, E3d-LSTM (Wang et al., 2018b) innovatively incorporates 3D convolutions into RNNs and introduces a gate-controlled self-attention module, thereby markedly improving long-term synthesis capabilities. Nonetheless, the increased computational complexity due to 3D convolutions might offset the performance gains in certain applications. MSPred (Villar-Corrales et al., 2022) proposes a hierarchical convolutional and recurrent network operating at multiple temporal frequencies to predict future video frames, as well as other representations such as poses or semantics.

**Challenges.** Recurrent networks, despite their effectiveness in capturing temporal dependencies, face several challenges in video prediction tasks. Their sequential nature, which allows them to model frame-by-frame changes, can lead to high computational complexity, especially in high-resolution scenarios. This is evident from the significantly higher FLOPs and lower FPS observed in recurrent-based models compared to their recurrent-free counterparts (Tan et al., 2023). Additionally, recurrent networks are prone to gradient vanishing and exploding issues, which can hinder their ability to learn long-term dependencies (Gao et al., 2022). These challenges underscore the need for alternative approaches that can balance efficiency and performance, such as recurrent-free models, which have shown promising results in various video prediction tasks.

### 3.1.2 Convolutional networks.

CNNs have been instrumental in the evolution of video synthesis technology. Beginning with GDL (Mathieu et al., 2015), the field has seen significant advancements. Following this, PredCNN (Xu et al., 2018) establishes a new benchmark by outperforming its predecessor PredRNN (Wang et al., 2017) across various datasets. SDC-Net (Reda et al., 2018) introduces a novel approach by utilizing a high-resolution video frame synthesis technique that effectively leverages past frames and optical flow. Building on these innovations, the introduction of SimVP (Gao et al., 2022) marks another milestone. This approach revisits the advancements made by ViT (Dosovitskiy et al., 2021) and introduces a simplified CNN network, demonstrating that such a configuration can achieve comparable performance in video synthesis.

**Challenges.** The CNN-based frame synthesis method, although simple to implement and fast, is not suitable for spatially shifting the pixels of the input frames. CAIN (Choi et al., 2020) and FLAVR (Kalluri et al., 2023) introduced channel attention and 3D-UNet in the synthesis of intermediate frames, respectively, but they do not completely replace explicit pixel motion methods such as kernel-based methods and flow-based methods. Moreover, in pursuit of efficiency, most CNN networks used for FFS have a relatively small number of parameters, generally not exceeding 60M (Tan et al., 2023). As a comparison, in order to leverage large datasets, the video diffusion model has been scaled up to over 1.5B parameters (Blattmann et al., 2023a). Scaling up CNN-based models effectively presents a significant challenge. We speculate that short-term prediction models for high-resolution real-time applications and models that attempt to draw knowledge and generation capabilities from large datasets will diverge.

### 3.1.3 Optical-flow-based synthesis.

The optical flow describes the motion of pixels between frames and can be used in flowing pixels of the current frame to synthesize near future frames (Liu et al., 2017). We consider the flow-based methods extensions of the kernel-based methods (Niklaus et al., 2017), as the kernel-based methods typically constrain the movement of pixels to a relatively small neighborhood (Cheng & Chen, 2021). Focusing on the enhancement of synthesis quality, FVS (Wu et al., 2020) utilizes a comprehensive approach by incorporat-

ing supplementary information such as semantic maps, instance maps, and optical flow from input frame sequences. This method, while effective, introduces challenges due to increased data modalities and computational demands. OPT (Wu et al., 2022b) estimates the optical flow of video motions in an optimization manner. By continuously refining the current optical flow estimation, the image quality of the next frame can be significantly improved. This approach effectively leverages the knowledge from off-the-shelf optical flow models (Teed & Deng, 2020) and frame interpolation models (Jiang et al., 2018; Huang et al., 2022b). Although training is omitted, the iterative optimization process during each inference requires substantial computation. DMVFN (Hu et al., 2023b) enhances the dense voxel flow (Liu et al., 2017) estimation by dynamically changing network architecture based on motion magnitude. DMVFN confirms the importance of a coarse-to-fine, multi-scale approach in solving short-term flows.

**Challenges.** The study of optical flow estimation is a hot topic in itself (Teed & Deng, 2020; Huang et al., 2022a; Sun et al., 2022; Dong & Fu, 2024). However, mainstream optical flow models trained on synthetic data with strong augmentations may differ significantly from the scenarios focused on by FFS. Moreover, the learning objective of these models is not to provide a flow that is suitable for moving pixels and synthesizing high-quality images. Xue et al. (2019) point out that for different downstream tasks, we need to fine-tune or even train a flow estimation network from scratch. Higher-performance optical flow networks may even lead to worse image synthesis effects because they may focus on ambiguous areas such as occlusions and have insufficient resolution (Niklaus & Liu, 2020; Huang et al., 2022b). In real-world scenarios, it is also challenging to obtain optical flow labels directly.

We believe that the near future frames can be synthesized increasingly better under the constantly improving optical flow method. And it is still a challenge to integrate optical flow methods into the goal of long-term video generation (Liang et al., 2023). Optical flow may only be used to predict pixel movement over a very short period of time and cannot help generate new video content.

### 3.1.4 Transformers.

After ViT's groundbreaking design of a pure transformer applied directly to sequences of image patches Dosovitskiy et al. (2021), the application of transformers to frame synthesis has become a hot topic. Video frame interpolation is a task very closely related to FFS (Liu et al., 2017). Both Shi et al. (2022) and Lu et al. (2022) propose transformer-based video interpolation frameworks to overcome the limitations of traditional CNNs, leveraging self-attention mechanisms to capture long-range dependencies and enhance content-awareness. They introduce innovative strategies, such as local attention in the spatial-temporal domain and cross-scale window-based attention, to improve performance and handle large motions effectively. Ye & Bilodeau (2023) present an efficient transformer model for video prediction, leveraging a novel local spatial-temporal separation attention mechanism, and compares three variants (fully auto-regressive, partially auto-regressive, and non-autoregressive) to achieve optimal performance and reduced complexity. There are still many ongoing studies aimed at advancing the use of transformers to address inter-frame dynamics with varying motion amplitudes and high-resolution synthesis issues (Park et al., 2023; Zhang et al., 2024).

**Challenges.** Many researchers believe that transformers have an advantage on large datasets (Zhai et al., 2022; Smith et al., 2023). However, most of these studies focus on high-level vision tasks. Moreover, the success of transformers on many tasks relies on fully leveraging the capabilities of foundational models. In image synthesis tasks, we are not yet certain about the best practices (Li et al., 2021). Leveraging the experience of large language models (LLMs) and similar architectures could be a promising direction, which we will discuss in subsequent Section 5.2.

## 3.2 Feature Space

Synthesizing in raw pixel space often overburdens models by requiring them to reconstruct images from scratch, a task particularly challenging for high-resolution video datasets. This realization has led to a shift in focus among researchers. Rather than grappling with the complexities of pixel-level synthesis, several studies have pivoted towards high-level feature synthesis in feature space, such as segmentation and depth maps. These approaches offer a more efficient way of handling the intricacies of videos (Oprea et al., 2020).

**Future semantic segmentation.** Future semantic segmentation represents a progressive approach of video synthesis, primarily focusing on synthesizing semantic maps for forthcoming video frames. This methodology diverges from traditional raw pixel forecasting, turning to semantic maps to narrow the synthesis scope and enrich scene comprehension. In this context, the S2S model (Luc et al., 2017) stands as a groundbreaking end-to-end system. It processes RGB frames alongside their semantic maps, both as input and output. This integration not only advances future semantic segmentation but also elevates the task of video frame prediction, showcasing the distinct advantage of semantic-level forecasting. Building on this foundation, SADM (Bei et al., 2021) further innovates by amalgamating optical flow with semantic maps. This fusion leverages optical flow for motion tracking and semantic maps for appearance detailing, employing the former to warp input images and the latter to inpaint occluded areas.

**Future depth prediction.** Depth maps, as a 2D data structure containing 3D information, can provide the model with an enhanced perception of the 3D world at minimal cost. Leveraging predictions of future depth maps can assist in FFS tasks. MAL (Liu et al., 2023) introduces a meta-learning framework with a two-branch architecture containing future depth prediction and an auxiliary task of image reconstruction. The proposed meta-learning framework improves the quality of synthesized future frames, especially in complex dynamic scenes.

**Challenges.** Future prediction in feature space, presents significant challenges due to the complex interplay of temporal dynamics and spatial context. Models must capture intricate motion patterns and predict accurate depths or semantic regions, requiring a deep understanding of 3D scene structures and object interactions. Ensuring temporal consistency and precise spatial accuracy while handling occlusions, perspective changes, and complex backgrounds is crucial. High-resolution feature maps and large-scale annotated datasets further increase computational demands and data requirements. Generalizing to unseen scenes and objects remains a formidable challenge, necessitating robust models that can adapt to diverse visual appearances and contexts. These challenges underscore the need for innovative approaches like meta-auxiliary learning to enhance future prediction capabilities.

## 4 Stochastic Synthesis

In the early stages, video synthesis is primarily perceived as a low-level computer vision task, with a focus on employing deterministic algorithms to enhance pixel-level metrics such as MSE, PSNR, and SSIM. This approach, however, inherently limits the potential for creative output from these models by confining possible motion outcomes to a single, fixed result (Oprea et al., 2020). This often results in high pixel-level scores but at the cost of producing blurry images, which significantly diminishes the practical applicability of these algorithms. Recognizing this issue, there has been a paradigm shift in the field of video synthesis, moving away from the reliance on short-term deterministic prediction towards embracing long-term stochastic generation. This transition acknowledges that while stochastic synthesis may yield results that diverge significantly from the ground truth, it plays a crucial role in fostering a more comprehensive understanding and enhancing creativity concerning the evolution of video content.

### 4.1 Stochasticity Modelling

Modeling the uncertain object motion can be achieved by introducing stochastic distributions into deterministic models, or directly leveraging probabilistic models.

**Stochastic distributions.** In the early stage, VPN (Kalchbrenner et al., 2017) uses CNNs for multiple predictions in videos based on pixel distributions and SV2P (Babaeizadeh et al., 2017) enhances an action-conditioned model (Finn et al., 2016) with stochastic distribution estimation for videos. Shifting the focus to a more holistic view of video elements, PFP model (Hu et al., 2020) presents a probabilistic method for synthesizing semantic segmentation, depth map, and optical flow in videos simultaneously. Additionally, SRVP (Franceschi et al., 2020) utilizes Ordinary Differential Equations (ODEs), while PhyDNet (Guen & Thome, 2020) employs Partial Differential Equations (PDEs) to calculate stochastic distributions. A

potential drawback is that their assumption that physical laws can be linearly disentangled from other factors of variation in the latent space, may not hold true for all types of videos.

**Probabilistic models.** With the pioneering work of GDL (Mathieu et al., 2015) establishing the foundation, adversarial training has significantly advanced FFS tasks in predicting uncertain object motions. Similarly, vRNN (Castrejon et al., 2019) and GHVAE (Wu et al., 2021) enhance VAEs through likelihood networks and hierarchical structures respectively, thereby offering another dimension to the ongoing evolution of stochastic synthesis methodologies.

Recognizing that object motion is largely deterministic barring unforeseen events like collisions, SVG (Denton & Fergus, 2018) models trajectory uncertainty using fixed and learnable priors, which effectively blends deterministic and probabilistic approaches. In a similar vein, but with a focus on enhancing temporal aspects, Retrospective Cycle GAN (Kwon & Park, 2019) introduces a sequence discriminator designed to detect fake frames. This idea of scrutinizing frame authenticity is further extended in DIGAN (Yu et al., 2022), where the emphasis shifts to a motion discriminator that concentrates on identifying unnatural motions.

To overcome pixel-level synthesis challenges in stochastic models, several works introduce intermediate representations. S2S (Luc et al., 2017) and Vid2Vid (Wang et al., 2018a) integrate adversarial training with future semantic segmentation. Additionally, TPK model (Walker et al., 2017) leverages a VAE to extract human pose information, and then a GAN for predicting future poses and frames.

It is worth noting that directly modeling stochastic distributions is prone to covering a broader predictive distribution but suffers from poor visual effects. In contrast, probabilistic models are capable of producing sharper results, yet they grapple with issues such as mode collapsing, training difficulties, and substantial computational overhead. Bridging these two methodologies, SAVP (Lee et al., 2018) combines stochastic distributions with adversarial training aiming to achieve broader predictive distributions with promising visual quality.

**Challenges.** Despite the ability of stochastic models to capture a broad spectrum of plausible futures, they often grapple with visual quality issues and heightened computational demands. Directly modeling stochastic distributions can result in blurred outputs, while probabilistic models may face challenges such as mode collapse and training instability. The quest for a harmonious balance between diverse, high-quality predictions and computational efficiency remains a formidable challenge. Moreover, the assumption that physical laws can be linearly disentangled from other variation factors may not universally apply to all video types, hinting at the need for more adaptable and generalizable models in the future.

### 4.2 Disentangling Components

Stochastic synthesis algorithms primarily focus on the randomness in motion. This approach, however, often neglects the birth-and-death phenomena of objects in videos. Consequently, numerous studies isolate motion from other video elements or artificially manipulate its evolution, providing a clearer understanding of motion dynamics while simplifying the complexity of real-world scenarios.

**Content and motion.** Video synthesis algorithms confront the challenging complexity of natural video sequences by emphasizing intricate image details. To this end, they focus on effectively modeling appearances through detailed local information, while simultaneously developing a comprehensive understanding of the dynamic global content inherent in videos. However, in applications such as robot navigation and autonomous driving, the importance of understanding object motion patterns supersedes the pursuit of visual aesthetics. This priority shift has encouraged the development of algorithms that emphasize predicting the movement of objects and distinguishing motion from appearance in videos. Early work CDNA (Finn et al., 2016) has set a precedent by explicitly predicting the motion of objects. It maintains invariant appearance characteristics, which is instrumental in applying the model to objects beyond those encountered during training.

MoCoGAN (Tulyakov et al., 2018) automatically learns to disentangle motion from content in an unsupervised manner, while the approach of leveraging separate content and motion encoder pathways has also been widely used in various video prediction models. This idea of separating content and motion is further explored

in LMC (Lee et al., 2021), which makes the motion encoder focused on motion prediction based on residual frames, while the content encoder extracts content features from the input frame sequence. MMVP (Zhong et al., 2023) takes a different approach by employing only one image encoder to extract information, and then a two-stream network before the image decoder to separately handle motion prediction and appearance maintenance.

Addressing the stochastic nature of motion, AMC-GAN (Jang et al., 2018) models multiple plausible outcomes via adversarial training. Transitioning to a different approach, SLAMP (Akan et al., 2021) adopts a non-adversarial approach but focuses on learning stochastic variables for separate content and motion. Further advancing this field, LEO (Wang et al., 2023e) and D-VDM (Shen et al., 2023) leverage diffusion models for more realistic disentanglement of content and motion, showcasing the latest advancements in this direction.

**Foreground and background.** In the process of predicting future frames, the motion dynamics of the foreground (objects) and background (scene) typically exhibit significant differences. Foreground objects typically exhibit more intense movement, whereas scenes tend to remain relatively static. This distinction has steered research towards predicting the motions of these elements separately, offering a more nuanced understanding of video dynamics. A pivotal contribution in this area is DrNet (Denton et al., 2017), which specifically addresses scenarios where the background remains largely unchanged across video frames. This model ingeniously decomposes images into the content and pose of objects, then harnesses adversarial training techniques to develop a scene discriminator that evaluates whether two pose vectors belong to the same video sequence. Similarly, OCVP-VP (Villar-Corrales et al., 2023) utilizes a slot-wise scene parsing network SAVi (Kipf et al., 2021) to achieve segmentation from the scene to the object level. Focusing on such videos, prediction models can streamline the prediction process by eliminating the need to learn complex scene dynamics. Human-centric tasks, such as predicting human movement and interaction with the environment, and object-centric tasks, such as tracking object motion and positioning, both benefit from this approach.

**Human-centric.** FFS often centers around foreground motion, particularly when it involves intricate human movements. In such scenarios, a common assumption across various specialized datasets is the relatively static nature of the background, which is a typical characteristic in datasets focusing on detailed human motion. This has led to a significant focus in research on understanding and forecasting human poses to enhance the prediction of foreground motion. An example of this approach is seen in the work of DVGPC (Cai et al., 2018), which innovatively predicts skeleton motion sequences before transforming these sequences into pixel space using a skeleton-to-image transformer. Their method is an effective solution for linking the abstract representation of motion with the practical aspect of video prediction, demonstrating a nuanced understanding of the complexities involved in human-centric FFS tasks.

**Object-centric.** The field of object-centric video prediction is first introduced in the work of CVP (Ye et al., 2019), which lays the foundation for this specialized area of video prediction. SlotFormer (Wu et al., 2022c) introduces transformer-based auto-regressive models to learn representations of each object in video sequences. This innovation ensures consistent and accurate tracking of each object over time. The OKID model (Comas et al., 2023) represents a recent advancement. It uniquely decomposes videos into distinct elements, specifically the attributes and trajectory dynamics of moving objects, employing a Koopman operator. This approach highlights a more detailed method of analyzing object motion within video sequences, differentiating it from previous methodologies.

**General.** Methods that concentrate on human poses or objects have demonstrated considerable promise in specific video datasets, yet they encounter limitations due to their reliance on pre-defined structures and struggle with adapting to variable backgrounds, which hampers their ability to generalize. This challenge is evident in their performance, which, while effective under certain conditions, falters when faced with dynamic background shifts, revealing a lack of the versatility required for wider applications. Bridging this gap, MOSO (Sun et al., 2023) emerges as a notable approach, identifying motion, scene, and object as the pivotal elements of a video. It delves deeper into content analysis by distinguishing between scene and object. Scene and object are considered as a further breakdown of content, where scene represents the background and object represents the foreground. MOSO's innovative contribution lies in its two-stage

network tailored for general video analysis. Initially, the MOSO-VQVAE model dissects video frames into token-level representations, honing its capabilities through a video reconstruction task. Subsequently, the model employs transformers in its second stage, tackling disparities in masked tokens. This strategic design equips the model to handle a variety of tasks at the token level, including video prediction, interpolation, and unconditional video generation.

**Challenges.** Disentangling content from motion or foreground from background in videos is complex due to the intricate interplay between temporal dynamics and spatial context. Models must accurately capture and predict motion patterns, depths, and semantic regions while maintaining temporal consistency and spatial accuracy. Handling occlusions, perspective changes, and complex backgrounds adds to the difficulty. High-resolution feature maps and large-scale annotated datasets exacerbate computational demands. Generalizing to unseen scenes and objects remains challenging, necessitating robust models that can adapt to diverse visual appearances and contexts. Applying the concept of separate processing to the generative methods we will discuss later (in Section 5) is also a challenge.

### 4.3 Motion-Controllable Synthesis

In the field of FFS, one specialized research direction has emerged that focuses on the explicit control of motion. This approach is distinct in its emphasis on forecasting future object positions based on user-defined instructions, diverging from the conventional reliance on past motion trends. The central challenge in this domain lies in the synthesis of videos that adhere to these direct instructions while maintaining a natural and coherent flow, a task that demands a nuanced understanding of both user intent and the dynamics of motion within a video context. This challenge highlights the intricate balance between user control and automated imagination, marking a significant shift in how FFS models are conceptualized and implemented.

**Strokes.** There is no historical motion information to be used for video generation from one still image, hence several methods have emerged allowing for interactive user control. iPOKE (Blattmann et al., 2021) introduces techniques where local interactive strokes and pokes enable users to deform objects in one still image to generate a sequence of video frames. These strokes indicate the user's intended motion for the objects. Following this innovative pathway, Controllable-Cinemagraphs model (Mahapatra & Kulkarni, 2022) introduces a method to interactively control the animation of fluid elements. The advancements in the field underscore the growing importance of user-centric approaches in the realm of motion-controllable FFS.

**Instructions.** The integration of instructions of various modalities, including local strokes, sketches, and texts, is increasingly common in works aiming to capture user-specified motion trends. VideoComposer (Wang et al., 2023a) synthesizes videos by combining text descriptions, hand-drawn strokes, and sketches. This approach respects textual, spatial, and temporal constraints, leveraging video latent diffusion models and motion vectors for explicit dynamic guidance. Essentially, It can generate videos that align with user-defined motion strokes and shape sketches. In a similar vein, DragNUWA (Yin et al., 2023) primarily leverages text for video content description and strokes for future motion control, enabling the generation of customizable videos. These attempts advance the field of video generation by broadening the range of user input modalities.

**Challenges.** Achieving natural and coherent video synthesis under explicit user control is challenging. Models must interpret user intent accurately and generate videos that adhere to specified motion instructions while maintaining temporal and spatial coherence. Balancing user control with the model's autonomous imagination is crucial. Ensuring that the generated videos are both visually appealing and contextually appropriate adds to the complexity, requiring a deep understanding of both user input and video dynamics.

## 5 Generative Synthesis

In video analysis, the focus shifts towards algorithms designed for generative video synthesis, especially when dealing with videos that exhibit stochastic birth-and-death events. These events introduce unpredictability

as objects come into existence and vanish. These algorithms necessitate a profound understanding of the underlying physical principles governing the real world to tackle such complexities. Instead of relying on simplistic linear motion predictions extrapolated from historical frames, they embrace the challenge with sophisticated and imaginative modeling techniques. As a result, tasks like transforming a single static image into a dynamic video, often referred to as the image animation problem, emerge as promising candidates for applying generative video prediction techniques.

## 5.1 Diffusion-Based Generation

Diffusion models (Ho et al., 2020) have emerged as the dominant approach in image generation. The Latent Diffusion Model (LDM) (Rombach et al., 2022) extends this capability into the latent space of images, significantly enhancing computational efficiency and reducing costs. This innovation has paved the way for the introduction of diffusion models into the realm of video generation.

**Latent diffusion model extensions.** Extensions of LDM have demonstrated robust generative capabilities in video generation (Voleti et al., 2022). For instance, Video LDM (Blattmann et al., 2023b) leverages pre-trained image models to generate videos, offering multi-modal, high-resolution, and long-term video generation capabilities. Similarly, SEINE (Chen et al., 2024a) introduces a versatile video diffusion model that creates transition sequences, enabling the generation of longer videos from shorter clips.

**Text-guided video completion with additional information.** Recent research efforts have focused on harnessing additional information along with RGB images to fulfill the text-guided video completion task. LFDM (Ni et al., 2023) extends latent diffusion models to synthesize optical flow sequences in latent space based on textual guidance. Seer (Gu et al., 2023) inflates Stable Diffusion (Rombach et al., 2022) along the temporal axis, allowing the model to use natural language instructions and reference frames to imagine multiple variations of future outcomes. Emu Video (Girdhar et al., 2023) generates an image conditioned on textual guidance and then extrapolates it into a video, making it flexible for creating videos based on different textual inputs. DynamiCrafter (Xing et al., 2023a) broadens the application of text-guided image animation to open-domain images. SparseCtrl (Guo et al., 2023) enables sketch-to-video generation, depth-to-video generation, and video prediction with an expanded input range. There are also methods, such as PEEKABOO (Jain et al., 2024), that attempt to achieve interactive synthesis, aiming to open up unprecedented applications and creativity.

**Preserving text guidance.** Some works aim to provide a more precise understanding of text guidance and preserve this information along the time dimension. MicroCinema (Wang et al., 2023d) adopts a divide-and-conquer strategy to address appearance and temporal coherence challenges. It uses a two-stage generation pipeline, initially creating the initial image using any existing text-to-image generator and then introducing a dedicated text-guided video generation framework for motion modeling. LivePhoto (Chen et al., 2023) introduces a framework that incorporates motion intensity as a supplementary factor to enhance control over desired motions. It also proposes a text re-weighting mechanism to emphasize motion descriptions, demonstrating impressive performance in text-guided video synthesis tasks. I2VGen-XL (Zhang et al., 2023) utilizes static images for semantic and qualitative guidance, showcasing diverse approaches in text-guided video synthesis research.

**Integrating autoregressive models.** Despite diffusion models becoming the most popular generative models, some researchers attempt to preserve the architecture of autoregressive models to simultaneously utilize the advantages of both mechanisms (Weng et al., 2023). Early generative video synthesis algorithms faced limitations in data availability and model scalability, but they laid the groundwork for integrating LLMs and diffusion models in video generation. Recently, GAIA-1 (Hu et al., 2023a) and Sora (Brooks et al., 2024) leverage the strengths of both diffusion models and LLMs to achieve more creative, general, and scalable video synthesis.

These advancements in video generation highlight the potential of diffusion models to create high-quality, controllable, and diverse video content, pushing the boundaries of what is possible in the field of computer vision and artificial intelligence.

**Challenges.** Diffusion models have made significant strides in video generation, yet several critical challenges persist. Maintaining temporal coherence and consistency across frames is essential for realism but remains difficult (Chen et al., 2024b; Xu et al., 2024). The computational efficiency and scalability of these models are hindered by their resource-intensive nature, limiting their widespread adoption (Peebles & Xie, 2023). Controllability and interpretability issues arise as textual guidance may not always align with visual outcomes, and model behaviors can be opaque. Data availability and diversity are crucial for training robust models, but obtaining comprehensive datasets is challenging.

## 5.2 Token-Based Generation

In the realm of image/video generation, diffusion-based methods have garnered significant attention. However, these algorithms typically possess smaller model sizes compared to contemporary large-scale language models (LLMs). Recent research has also placed significant emphasis on exploring how LLMs can accomplish these tasks, leveraging extensive related research and optimization methods from LLMs to validate scaling laws in the visual domain.

**Key components.** Implementing transformers FFS involves two crucial elements: an efficiently scalable LLM framework and a proficient image/video tokenizer. Innovations such as VQ-VAE (Van Den Oord et al., 2017) and VQGAN (Esser et al., 2021) have combined auto-regressive models with adversarial training strategies to address image quantization and tokenization. An effective visual tokenizer should minimize tokens per image or video clip segment while ensuring near-lossless visual reconstruction. However, the high token requirements for lossless reconstruction, especially for high-resolution images, pose challenges for processing long video sequences during training, limiting video generation capabilities.

**Early transformer applications.** Before the advent of LLMs (Brown, 2020; Achiam et al., 2023), transformers have already made a significant impact on time-series modeling. Video Transformer (Weissenborn et al., 2019) pioneers the application of transformer architectures in video synthesis, creating an auto-regressive model. Despite its success, it inherits the common drawbacks of transformers: extensive training resource consumption and prolonged inference time.

**Latent space modeling.** The Latent Video Transformer (LVT) (Rakhimov et al., 2020) introduces a novel approach by modeling dynamics and predicting future features auto-regressively within a latent space, reducing the computational burden. The NUWA framework (Wu et al., 2022a) proposes a versatile 3D transformer encoder-decoder architecture adaptable to various data modalities and tasks, further showcasing the potential of transformers in video synthesis. NUWA-Infinity (Liang et al., 2022) expands on this with an innovative generation mechanism aimed at achieving infinite high-resolution video generation, reflecting ongoing efforts to unify generative tasks across different modalities.

**Sequential modeling and visual sentences.** LVM (Bai et al., 2023) introduces sequential modeling to enhance the learning capabilities of large-scale vision models, demonstrating the scalability and flexibility of sequence models in in-context learning. The concept of "visual sentence" was proposed, where a sequence of images with intrinsic relationships is arranged similarly to a sentence in the language, enabling the model to leverage sequence information for sentence continuation and other visual tasks without relying on non-pixel-level knowledge.

**In-context learning in visual domain.** In-context learning is not new to the visual domain. Painter (Wang et al., 2023b) introduces a general framework for visual learning that enables images to "speak" through in-context visual learning, enhancing both image generation and understanding. Seg-GPT (Wang et al., 2023c) investigates using a GPT architecture for image segmentation, proposing the concept of segmenting everything and demonstrating the possibility of achieving unified image segmentation in an unsupervised learning environment, advancing generalization in visual segmentation tasks.

**Advances in video generation.** In the video generation domain, MAGVIT (Yu et al., 2023a) presents a masked generative video Transformer that efficiently processes video by masking certain parts and generating

missing segments. MAGVIT-v2 (Yu et al., 2023b) suggests that transformers might outperform diffusion models in visual generation tasks, emphasizing the critical role of visual tokenizers. VideoPoet (Kondratyuk et al., 2023) introduces a large language model for zero-shot video generation, pushing the boundaries of unsupervised learning in video generation. It allows users to generate or modify videos based on high-level textual prompts, excelling in capturing temporal and contextual relationships within video data.

Text-guided generative video synthesis algorithms create a sequence of frames by integrating context frames and textual guidance. The Text-guided Video Completion (TVC) task involves completing videos based on various conditions, including the first frame (video prediction task), the last frame (video rewind task), or a combination of both (video transition task), guided by textual instructions. MMVG (Fu et al., 2023) addresses the TVC task by utilizing auto-regressive encoder-decoder architectures, integrating text and frame features, resulting in a unified framework capable of handling multiple video synthesis tasks.

These studies collectively integrate visual tokenizers and large language models to offer unified and scalable frameworks for visual learning, driving breakthroughs in FFS tasks. The evolving landscape of FFS research continues to showcase the potential of transformers and LLMs in unifying generative tasks across different modalities.

**Challenges.** Token-based generation for FFS faces significant hurdles, including efficient visual tokenizer to balance minimal tokens with near-lossless reconstruction, particularly for high-resolution content. The computational resource demands of transformer-based models pose adoption barriers, especially in resource-constrained settings. Because when the computing power budget is insufficient, we may not observe the emergence phenomenon reported in previous studies under this paradigm (Bai et al., 2024). Recently, Sun et al. (2024a) suggest that one reason token-based models cannot match the visual quality of diffusion models is the lack of integration of high-quality community assets, such as training infrastructure and high-quality data.

# 6 Application Realms

The applications of FFS span a wide range of domains, showcasing its significance in different fields.

**World model.** World models (Ha & Schmidhuber, 2018; Zhu et al., 2024) provide a more general framework for simulating and predicting the behavior of complex systems. A world model is often used in reinforcement learning and robotics to enable agents to make informed decisions and take actions that lead to desired outcomes. To construct a world model, FFS is an important learning objective (Hafner et al., 2020; 2023a; Wang et al., 2024a). Escontrela et al. (2024) show that video prediction can also serve as a part of reward modeling to aid reinforcement learning.

**Autonomous driving.** FFS is indispensable for autonomous vehicles, including self-driving cars and drones, as it allows them to anticipate the movement of objects, pedestrians, and other vehicles. This capability is crucial for ensuring safe navigation. For instance, GAIA-1 (Hu et al., 2023a) leverages a unified world model that integrates multi-modal large language models and diffusion processes to predict control signals and future frames, enhancing the vehicle's decision-making capabilities. Currently, how to effectively utilize image information to aid trajectory prediction in real-world scenarios remains a challenge (Nayakanti et al., 2023; Varadarajan et al., 2022).

**Robot navigation.** In robotics, FFS is employed to guide robots through dynamic environments. It enables robots to plan their paths, move objects, and avoid obstacles effectively, as demonstrated by Finn & Levine (2017). By predicting future states, robots can make proactive decisions, enhancing their adaptability and efficiency in complex settings.

**Cinema industry.** FFS finds application in the cinema industry, particularly in special effects, animation, and pre-visualization. It aids filmmakers in creating realistic scenes and enhancing the overall cinematic experience. For example, Mahapatra & Kulkarni (2022) utilize FFS to generate visually compelling sequences that contribute to the storytelling and artistic expression in films.

**Meteorological community.** FFS plays a crucial role in weather forecasting, aiding meteorologists in simulating and predicting atmospheric conditions. By accurately predicting future frames, FFS contributes to improved accuracy in weather predictions, as exemplified by the work of Shi et al. (2017). This capability is vital for weather forecasting services and disaster preparedness.

**Anomaly detection.** Liu et al. (2018) propose an approach to anomaly detection in videos by leveraging future frame prediction, where normal events are expected to be predictable while abnormal ones are not. The method introduces a motion constraint in addition to appearance constraints to ensure the predicted future frames are consistent with the ground truth in both spatial and temporal aspects.

Overall, these diverse applications underscore the importance and potential of FFS as a tool for understanding and interacting with the world around us. Its ability to predict future states from past observations makes it a valuable asset in fields ranging from artificial intelligence to entertainment and beyond.

**Challenges.** When migrating the FFS methods to applications, potential issues that may be encountered. For instance, in specific domains, we might need to explore solutions for few-shot learning (Gui et al., 2018) or adaptation during inference (Choi et al., 2021). We also need to study how to reasonably integrate vectorized data with image data. Moreover, in real-world tasks, people often prefer methods with good interpretability, while most end-to-end FFS methods are not easily interpretable.

# 7 Related Work

**Previous Surveys on Video Prediction.** The field of video prediction and related topics, such as action recognition and spatio-temporal predictive learning, has seen significant advancements in recent years, largely driven by deep learning techniques. Several comprehensive surveys have provided an overview of the state-of-the-art methods, datasets, and evaluation metrics in this domain. Zhou et al. (2020) introduce next-frame prediction networks before 2020, categorizing them into sequence-to-one and sequence-to-sequence architectures. The paper compares these approaches by analyzing network architecture and loss function design, providing a quantitative performance comparison using off-the-shelf datasets and evaluation metrics. Oprea et al. (2020) offer a review of deep learning methods for video prediction, defining the fundamentals and background concepts, and analyzing existing models according to a proposed taxonomy. The survey includes experimental results to facilitate the assessment of the state of the art on a quantitative basis. Rasouli (2020b) provide an overview of vision-based prediction algorithms, focusing on deep learning approaches. The paper categorizes these algorithms into video prediction, action prediction, trajectory prediction, body motion prediction, and other prediction applications, discussing common architectures, training methods, data types, evaluation metrics, and datasets. Kong & Fu (2022) survey the state-of-the-art techniques in action recognition and prediction, discussing existing models, popular algorithms, technical difficulties, action databases, evaluation protocols, and future directions. Tan et al. (2023) propose OpenSTL, a comprehensive benchmark for spatio-temporal predictive learning, categorizing approaches into recurrent-based and recurrent-free models. The paper conducts standard evaluations on various datasets and provides an analysis of how model architecture and dataset properties affect performance.

**Surveys on Video Diffusion Models.** Xing et al. (2023b) present a comprehensive review of video diffusion models in the AIGC era, categorizing the work into video generation, video editing, and other video understanding tasks. The survey includes a thorough review of the literature in these areas and discusses challenges and future trends. Li et al. (2024) present the survey of recent advancements in long video generation, summarizing them into two key paradigms: divide and conquer, and temporal auto-regressive. It also provides a comprehensive overview and classification of datasets and evaluation metrics, discussing emerging challenges and future directions in this dynamic field. Sun et al. (2024b) review Sora, OpenAI's text-to-video model, categorizing literature into evolutionary generators, excellent pursuit, and realistic panorama, while discussing datasets, metrics, challenges, and future research directions.

**Our Survey Focus.** Our survey comprehensively reviews historical and contemporary works in FFS, focusing on the transition from deterministic to generative synthesis methodologies. The survey highlights significant advancements and shifts in approach, emphasizing the growing importance of generative models in achieving realistic and diverse future frame predictions.

## 8 Conclusion

In this survey, we have discussed various aspects of FFS, covering key topics such as predominant datasets, evolving algorithms and current challenges.

Based on the overall development trends in the field of artificial intelligence, we believe that the trajectory of video synthesis research should diverge. On one hand, research should focus on the trend towards model lightweighting suitable for high-definition video applications, targeting low-level objectives such as video compression and short-term motion estimation. On the other hand, research should explore how models can fundamentally understand the laws of the world and create content by harnessing substantial computational resources and diverse, long-duration video data. For the latter, future research should prioritize the formulation of evaluation metrics that incentivize stochastic synthesis, thus expanding the potential of simulation of the human world. The ultimate objective is to cultivate video synthesis models with a profound comprehension of the inherent dynamics within videos. Such models could execute video generation over extended time horizons with high stochastic complexity in the real world.

Our taxonomy is based on algorithm stochasticity, showcasing a significant shift from deterministic approaches toward generative methodologies. This survey emphasizes the importance of balancing pixel-level accuracy with a deep understanding of complex scene dynamics in video synthesis. Additionally, we delve into the intricacies of stochastic birth-and-death phenomena, advocating for enhanced evaluation metrics and the utilization of substantial computational resources and large-scale video datasets. We also sort out the different research directions under this topic and discuss the current challenges. These insights aim to guide future research directions in video synthesis. As the field advances, we anticipate the development of models that provide a more profound and nuanced grasp of nature in the real world. This evolution promises to enhance accuracy, efficiency, and creativity, paving the way for novel applications and research.

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
