# OpenReview forum: "A Survey on Future Frame Synthesis: Bridging Deterministic and Generative Approaches"
_TMLR — Withdrawn by Authors_

### Review · Reviewer_oJ3W · 2024-12-29

**Summary Of Contributions:**

The paper provides a survey of different learning-based approaches for video prediction. Specifically, the paper categorizes two major types of future frame synthesis models: deterministic models and stochastic models, and discuss representative methods within each catgories.

**Audience:**

Yes

**Claims And Evidence:**

No

**Requested Changes:**

See the weakness section above:

* Highlight different architectural design of video synthesis models. This could include key components, design philosophies, or innovative features that distinguish one approach from another.
* An overview of different models' capability or empirical performance.

**Strengths And Weaknesses:**

Strengths:

* The paper provides a comprehensive review of different future frame synthesis methods, along with the motivation and challenges of each of them.

* The paper establishes clear and logical connections between different approaches.

Weaknesses:

* The paper lacks detailed methodological and architectural descriptions of the models it introduces. At present, most models are described only through brief textual overviews. It would be helpful to provide more in-depth explanations, such as diagrams of what each representative models is capable of and incapable of, or key architectural components. Furthermore, the paper provides limited discussion on the empirical performance of the various models reviewed. While it is understandable that presenting unified benchmark results is challenging—given that the models target diverse tasks such as semantic segmentation, optical flow prediction, and more—an analysis or comparison of performance trends across these tasks would have been valuable. Highlighting common evaluation metrics, datasets, or task-specific benchmarks could offer readers a clearer understanding of the strengths and weaknesses of each approach within its respective context.

* The writing and presentation of certain parts can be improved.  For example, in equation 1&2, is X, Y the RGB value of pixels? It is unclear to me why synthesizing new frame Y amounts to multiplying pixel values X with a probability density. In section 4, the paper introduces some GAN and VAE models, labeling them as stochastic approaches, then in section 5 the paper introduces diffusion and autoregressive models and labels them as generative approaches, however at its essence, these models are very similar, for instance we can estimate likelihood via  VAE, diffusion or AR models; we can do video2video task using either GAN or diffusion.

---

### Review · Reviewer_Ue4H · 2025-01-06

**Summary Of Contributions:**

This paper surveys deterministic and generative approaches to future frame synthesis (FFS) in computer vision. The authors taxonomize the FFS task into referencing pixels and generating new frames. Following a brief introduction to the FFS task, the authors cover datasets often used in FFS and the main challenges from a dataset perspective. The paper covers deterministic synthesis, stochastic synthesis, and generative synthesis. Finally, the authors discuss several application domains and related work before concluding the survey.

**Audience:**

No

**Claims And Evidence:**

Yes

**Requested Changes:**

- The "birth-and-death" notion can be explained in more detail
- From an organizational perspective, I think it would make more sense to discuss the main approaches first and follow it up with datasets, evaluation, metrics, and challenges.
- It would be good to add much more detail and analysis of the methods discussed in the deterministic, stochastic, and generative categories.
- It would be beneficial to add a couple of paragraphs to compare the different methods
- The applications section also seems a bit abridged. It would make the paper stronger to elaborate on each of the applications.

**Strengths And Weaknesses:**

## Strengths
- The survey highlights several important challenges faced in the FFS literature
- The proposed taxonomy is generally sound

## Weaknesses
- This survey seems to be quite limited in scope and does not seem to provide any new perspectives for the FFS task.
- The paper covers a few papers in each category but does not provide a comprehensive summary of these works and their advantages and disadvantages compared with other works.
- Overall, the three approaches: deterministic, stochastic, and generative are not described in detail and the paper assumes the audience already has knowledge of these paradigms.
- There is a lot of redundancy in this paper with references to generalization, blurry outputs, and computational demands of the methods.

---

### Review · Reviewer_Da67 · 2025-01-28

**Summary Of Contributions:**

Video Generation is at a pivotal moment today with tremendous progress being made in just a year. A detailed survey outlining the history  is apt in terms of timing. The paper attempts to capture all the developments in this area highlighting the challenges, drawbacks, and benefits of each approach. It starts with a short introduction of the several datasets that have been adopted by the community and delves into the progression that led to the models that we have today. The paper concludes with a brief discussion around potential applications.

**Audience:**

Yes

**Broader Impact Concerns:**

I don't have any concerns.

**Claims And Evidence:**

Yes

**Requested Changes:**

1. Bring more structure to the paper,
suggestion 1: provide a quick summary of the different categories as a section before delving into each category in detail sequentially.
suggestion 2: create a table that lists all the models and the meta data in the columns (dataset used, publication date, category, modeling technique, remarks).
suggestion 3: add some results, better if the results are a comparison between different models. What would be most insightful for me is to be able to see how to these models compare, and where do they lack **visually**. No need to compare all possible models, but at least one representative model from each category and the distinction between the different techniques visually that the paper tries to highlight now via text.

2. Involve a detailed discussion around how data impacts all these models and how well do they generalize.

3. Improve the coverage, add papers that might be missing (StyleGAN-V, INR-V, LLaVA, etc.)

4. Include an extensive discussion around today's progress and juxtapose against the earlier models.

**Strengths And Weaknesses:**

Strenghts:
1. The paper is very detailed and covers a broad range of modeling techniques.
2. It does a good job of outlining the challenges and benefits of each technique and the evolution to the next set of models.
3. The paper could be a good starting point for a newcomer to the field.


Weakness:
1. The paper severely lacks structure. As a regular contributor to the field, I found the paper to be monotonous and a long summary of the existing works. Albeit, that is the intention of a survey paper, a visualization of some sort would greatly help the reader. For example, a page long table that divides all the mention paper into a given category and also outlines the contribution of each model--this could be a quick reference for a regular contributor and could greatly boost the paper.
2. Although the paper does a great job of outlining each paper, I would also like to see what dataset each of these models were trained on? Video generation is a highly data-driven field and I would be very interested on the implications of data size, generalization to scenarios outside the dataset, and how data has impacted this field. The paper briefly introduces the different datasets at the start, but there is no conversation around the implications of these datasets.
3. Although very comprehensive, the paper does miss citations. Two of the papers I can think of are "StyleGAN-V: A Continuous Video Generator with the Price, Image Quality and Perks of StyleGAN2 Ivan Skorokhodov, Sergey Tulyakov, Mohamed Elhoseiny" and "INR-V: A Continuous Representation Space for Video-based Generative Tasks Bipasha Sen, Aditya Agarwal, Vinay P Namboodiri, C. V. Jawahar". I would suggest a more thorough coverage.
4. **Missing discussion around SOTA video generation models**: The paper provides a historical overview of the video generation models but misses a transition to today's SOTA models (both paid & unpaid and open-source models) like LLaVA, Luma AI Labs, Pika, which is very important in my view. There needs to be an open discussion around how the models have evolved historically and juxtapose it with the advancements that we see today. I would like to see an extensive discussion around today's technology and the insights behind why we see the developments today and what made it possible (e.g. GPU? data? open-sharing? models?)

---

### Author Response · Authors · 2025-07-14

This paper was accepted after being resubmitted. Thanks to the relevant reviewers, editors, and organizers!

https://openreview.net/forum?id=ZN4rzrHlNz

---

### Note · Authors · 2025-02-14

**Comment:**

We plan to make substantial updates and resubmit. Thank you to all the reviewers.

**Withdrawal Confirmation:**

I have read and agree with the venue's withdrawal policy on behalf of myself and my co-authors.